# Joint Fusion and Detection via Deep Learning in UAV-Borne Multispectral Sensing of Scatterable Landmine [note 1]

**DOI:** 10.3390/s23125693

**Published:** 2023-06-18

**Authors:** Zhongze Qiu, Hangfu Guo, Jun Hu, Hejun Jiang, Chaopeng Luo

**Affiliations:** 1School of Electronics and Communication Engineering, Sun Yat-sen University, Shenzhen 518107, China; 2Science and Technology on Near-Surface Detection Laboratory, Wuxi 214035, China

**Keywords:** deep learning, landmine detection, UAV-borne, multispectral fusion, object occlusion

## Abstract

Compared with traditional mine detection methods, UAV-based measures are more suitable for the rapid detection of large areas of scatterable landmines, and a multispectral fusion strategy based on a deep learning model is proposed to facilitate mine detection. Using the UAV-borne multispectral cruise platform, we establish a multispectral dataset of scatterable mines, with mine-spreading areas of the ground vegetation considered. In order to achieve the robust detection of occluded landmines, first, we employ an active learning strategy to refine the labeling of the multispectral dataset. Then, we propose an image fusion architecture driven by detection, in which we use YOLOv5 for the detection part, to improve the detection performance instructively while enhancing the quality of the fused image. Specifically, a simple and lightweight fusion network is designed to sufficiently aggregate texture details and semantic information of the source images and obtain a higher fusion speed. Moreover, we leverage detection loss as well as a joint-training algorithm to allow the semantic information to dynamically flow back into the fusion network. Extensive qualitative and quantitative experiments demonstrate that the detection-driven fusion (DDF) that we propose can effectively increase the recall rate, especially for occluded landmines, and verify the feasibility of multispectral data through reasonable processing.

## 1. Introduction

Landmines have caused great harm to human life and health and greatly hindered the harmonious development of human society [1]. According to incomplete statistics, more than 110 million landmines are still present in post-war areas. To date, many methods have been developed to detect and remove landmines during war or post-war reconstruction. They can be divided into two categories based on their fundamental principles [2]. One employs the geophysical differences between certain parts of landmines and their surroundings, including ground-penetrating radar, low-frequency electromagnetic induction, metal detectors, ultrasonic devices, infrared imaging, and other combinations of detection techniques. The other has a wide range of research dimensions based on particular techniques, such as acoustics, biological sensing, and molecular tracking.

With the development of military technologies, scatterable landmines have emerged and are attracted increasing attention, as opposed to landmines covered with vegetation. The scatterable landmines naturally possess the advantages of speed, flexibility, and high mobility. Consequently, large-scale and large-area minefields can be deployed quickly during war. Moreover, aerial minelaying or projection from armored vehicles results in the randomized distribution of landmines in diverse topographies and complex vegetation environments. These pose new challenges for minefield clearance and make the long-distance and rapid detection of scatterable landmines increasingly necessary.

Meanwhile, massive advances in unmanned aerial vehicles (UAVs) have presented enormous potential [3]. Specifically, UAVs with various on-board electromagnetic sensors have yielded a considerable number of applications, including a rise in low-altitude landmine detection, as they can provide better safety performance and faster measurements than other mine detection platforms. Many investigations aiming at the combination of drones and predominantly ground-penetrating radar (GPR) have been conducted [4,5,6,7,8,9]. The authors in [4,6] aimed to achieve a precise and steady flight over the surface, to increase the positioning accuracy and reliability of GPR measurements, while on-board software-defined radio (SDR)-based radars were developed in [6,10] to detect landmines. Other customized sensors mounted on drones have also emerged. Among them, synthetic aperture radar (SAR) [9,11] and hyper-spectral [12] imaging techniques have shown unprecedented potential with advanced image processing and computer vision algorithms. In [11], the authors reported the capability of high-resolution 3D SAR images for the detection of buried hazards, while the hyper-spectral imaging of landmines was highlighted with different signal processing techniques in [12,13].

Numerous factors, such as the materials and shapes of the mines, terrain complexity, vegetation humidity, flight altitude, etc., limit the detection of scatterable mines. It is challenging for a single sensor to solve all of the aforementioned issues. It is well known that objects exhibit different features in different spectra, and multispectral remote sensing that obtains and comprehensively exploits different spectra information of the same target by means of photography or scanning is appealing. For instance, a particular spectrum may offer a unique advantage in certain terrains or settings, whereas the proper fusion of multispectral remote sensing might expand the observation range and reveal more information. Multispectral fusion greatly outperforms a single spectrum in pedestrian recognition, especially in challenging situations such as occluded appearances and background clutter [14]. However, the development of multispectral techniques in the detection of scattered landmines is still in a preliminary exploration stage. Research in this area mainly concentrates on the analysis of different spectral signatures and the simulation of ground environmental influences [15,16,17,18,19]. Differences in spectral signatures between soils and landmines are essential for detection, and a study has been performed regarding this [15]. The utilization of remote sensing thermal infrared technology has led to satisfactory results in PFM-1 minefield detection [16], and plastic mines such as PFM-1 have higher thermal conductivity when the heat flux is high in the early morning [18]. In addition, ref. [17] simulated the influence of environmental forces on the orientation of PFM-1, complicating the minefield environment.

In the past few years, deep learning, based on the cognitive structure of the human brain, has achieved great success in many areas. Compared with the traditional machine learning algorithm (shallow learning), it can extract more abstract features through multi-layer nonlinear transformation. As one of the fundamental visual tasks, object detection based on deep learning has outperformed traditional methods over the last few years. Ideas and implementations based on convolutional neural networks (CNNs) have facilitated the success of deep learning in detection [20]. Widely acclaimed deep learning models, such as ResNet [21], regions with CNN features [22] (R-CNN), Faster R-CNN [23], etc., show outstanding transferability in mine detection scenes [19,24,25,26]. Some customized deep learning models also show powerful feature extraction abilities and adaptability in varied scenes [27,28]. It is worth noting that different types of data were used in the above studies, such as thermal infrared images, GPR B-scans, and radar signals, but deep learning methods still achieve excellent performance.

The Binghamton University team [16,17,18,19] is conducting ongoing research into multispectral landmine detection. Comparative experiments indicate that thermal cameras can detect temperature variations between plastic PFM-1 landmines and host geology during the early-morning hours, when the thermal flux is high [18]. Their preliminary studies in [16,17] demonstrate that a thermal camera mounted on a low-cost UAV can successfully detect PFM-1 mines at the elevation of 10 meters above the ground, regardless of their orientations. In 2020, they incorporated an analysis of multispectral and thermal datasets and discovered that different bands are beneficial in different situations [19]. Furthermore, they employed the Faster R-CNN [23] model and yielded satisfactory accuracy for a partially or wholly withheld testing set. The employed data only consist of RGB visible images, without fusing other bands for detection, but their solid work gives confidence and inspiration for multispectral detection. In our past work [29], we combined the detection results of RGB and NIR images and finally achieved excellent performance, demonstrating the advantages of multispectral over monospectral images in detection. In this paper, to further investigate the application of multispectral imaging, we explore in depth a new approach as an extension of the aforementioned work.

Most of the existing research and trials only consider an extremely elementary simulation environment and landmine setting. However, high-quality data from the real environment or diversified and high-fidelity experimental scenes are the foundation for the sufficient generalization of supervised deep learning. Therefore, in this study, we were motivated to set up complex minefield conditions to simulate real scenes and collect raw high-quality multispectral data acquired via a drone with an integrated multispectral camera. One RGB sensor and four monochrome sensors collected data in discrete spectral bands—green, red, red edge, and near-infrared. More specifically, our dataset contains four types of scatterable landmines, four scenes with different ground vegetation, and various flight altitudes. To overcome the labeling issues introduced by the severe occlusion and the tiny size of the landmines and improve the labeling efficiency, we utilize the idea of active learning to ensure the complete and accurate labeling of the dataset. To improve the detection performance for the obtained high-quality multispectral images, we present a detection-driven image fusion framework with YOLOv5 [30] plugged in. When compared to the original YOLOv5 [30] benchmark or other fusion detection cascade models, our proposed fusion detection enhances the recall markedly. The RGB and NIR bands are used to demonstrate this in the study. There are 789 pairs of images in our dataset, which are partitioned into training and test sets in an 8:2 ratio. A total of 991 mines are labeled in the test dataset, and a recall rate of 0.848 is achieved when using RGB alone for prediction, while 0.906 is achieved with the fusion of NIR and RGB driven by the landmine detection task. As a plug-and-play module, it can be integrated into most of the current state-of-the-art object detection models.

The rest of the paper is organized as follows. Section 2 illustrates the details of our dataset, including the instruments and the data processing. Section 3 introduces the adopted deep learning method and proposed fusion module. Section 4 presents the experimental results and comparative analysis. Finally, conclusions are drawn in Section 5.

## 2. Multispectral Dataset

In order to address the shortage of public multispectral landmine datasets, we constructed a small landmine dataset, including labeled RGB and near-infrared (NIR) image pairs.

### 2.1. Landmine Samples

In the dataset, 40 mines with five different sizes and materials were selected in total, three of which were anti-personnel mines, while the others were anti-vehicle mines. Their primary materials were metal, and the shell parts contained a small amount of plastic and rubber. The specifications of the four types of mines are given in Figure 1.

With the exception of the M93 landmine, they are all olive-drab-covered and cylindrical in shape. Specifically, the smallest anti-personnel mine M14 is less than 6 cm in diameter and primarily non-metallic. In comparison, the anti-vehicle M93 Hornet mine possesses a relatively large body size and is plastic-cased with a rubber top. Before the explosion, they are air-delivered and hidden well on the ground or in vegetation, remaining active without being noticed for long periods.

### 2.2. Equipment

We acquired the multispectral images using the Sequoia multispectral kit. The Sequoia sensor comprises four monochrome cameras with specific narrow-band filters and high-resolution RGB lenses in the NIR and visible domains, respectively, as shown in Table 1.

The camera is equipped with a “sunlight sensor” that measures sky irradiance. The built-in spectroradiometer has precision of 1.5 nm to measure irradiance of 2048 pixels in the visible and NIR spectral ranges (350–1000 nm). The drone equipped with the multispectral sensor (Figure 2) can sustain a continuous flight of an hour approximately.

The preparations before take-off include the calibration of the drones and multispectral cameras and radiation correction through the sunshine sensor. The matching sunshine sensor can monitor radiation fluctuations to obtain multispectral images with stable quality under different lighting and natural conditions. The flying height changes according to the host environment and whether sparse, tall plants are present. In this work, the flying heights are fixed at approximately 3 m and 8 m, respectively, to obtain images with a similar ground resolution. During the data acquisition process, the drone cruises at a constant speed with an area overlap rate of 80% to ensure complete coverage of the minefields of the given range. Moreover, the built-in GPS/GNSS of the multispectral camera can determine the device’s location when capturing images, which provides good feasibility to convert the image coordinate system and the geographic coordinate system.

### 2.3. Experimental Scenes

We conducted data collection experiments and placed dozens of mines on grassland, instead of experimental scenes such as sand land, cement roads, and gravel roads, etc. A more complex environment with vegetation is considered to validate the feasibility of our method. We selected several low-vegetation environments to simulate minefields. As displayed in Figure 3, mines are easily obscured by wild vegetation. The mines were scattered randomly, with approximately 60% obscured by vegetation; in other words, only part of the mine body was visible. Moreover, a small number of them were entirely covered by ground vegetation, which caused extremely low visibility. In addition, there were many leaves, stones, and plant rhizomes that were similar in color and shape to the mines, which may have caused false alarms.

### 2.4. Acquisition and Registration

With a constant capturing interval of 2 s, UAVs may complete the minefield cruise in a very short amount of time. Sequoia’s monochrome and RGB cameras are not calibrated throughout production, resulting in a lack of rigorous time synchronization due to the various shutters. As a result, the collected multispectral raw images have a spatio-temporal shift. Thus, in addition to the typical nonlinear intensity differences, there are also spatio-temporal mismatches between the multi-modal data. For instance, the vegetation in the simulated landscape is mostly filled with dense grasses that sway in the wind in different directions. Completely accurate image registration is almost impossible. This can be a significant problem for actual minefield detection, but not for host environments without vegetation. Registration allows image pixels to be aligned for more precise fusion and also eases subsequent labeling. The registered infrared and visible image pair that we obtained is shown in Figure 4. We also refer to the channel feature of oriented gradients [31] (CFOG) method to register the multispectral images, which could effectively further improve the precision of the registration result. By comparing the positions of the corresponding landmines, we could directly assess the quality of the registration result.

### 2.5. Labeling

Supervised learning strongly relies on high-quality labeled data, which usually comes with high costs. Manually labeled data are time-consuming to obtain and sometimes unreliable. Even in the well-known public datasets in the academic community, there are still many instances of mislabeling. In this paper, we labeled our dataset using the following steps: detection equipment assembly, data collection, data cleaning, image alignment, and labeling.

Firstly, random landmine placement and extreme occlusion substantially increase the difficulty of labeling. Moreover, nonsalient objects such as landmines usually blend in easily with their surroundings. In a 1280 × 960 resolution image, the target is estimated to occupy 100 pixels, while, for the smallest mine, M14, it occupies only approximately 10 pixels. It is not feasible to label each image manually. Therefore, during the labeling precess, we referred to the strategy described in [14]. We compared and labeled images in parallel after aligning the image pairs. Moreover, to accelerate the labeling, we introduced an active learning technique, which is shown in Figure 5.

In real-world circumstances, it is worth considering how to collect valuable labeled data for a lower cost and subsequently increase the algorithm’s performance. Active learning focuses on reducing the burden of labeling by incorporating human experience into machine learning models. Firstly, the query function determines the candidate set of data to be labeled. Then, professionals with business experience annotate the selected datasets. With continual manual annotation to be added, the model is therefore incrementally updated. The core principle behind active learning is to manually resolve the “hard sample” via human interference. In our situation, the term “hard sample” refers to landmines that are well hidden and difficult to detect. Furthermore, the training results assist in the improvement of the annotation. After three rounds of the iterative process, a dataset with reliable annotations was created.

## 3. Methods

Existing image fusion algorithms tend to focus on the visual effect of the fused image itself and the pursuit of quantitative evaluation, while ignoring the promotion ability of image fusion as a low-level vision task for the subsequent high-level vision task. In fact, the ideal image fusion method should not only provide rich texture details and accord with human visual perception, but also help to conduct high-quality downstream tasks.

Compared with the traditional architectures, the fusion architecture driven by high-level vision tasks can better conserve semantic information as well as exclude external noise. Liu in [32] designed a cascade structure to connect the image denoising module with the high-level vision network and jointly trained them to minimize the image reconstruction loss. Under the guidance of semantic information, a denoising network could further improve the visual effect and help to complete the semantic task. Similarly, Guo in [33] designed a residual network for semantic refinement and jointly trained the “drainage” network and the segmentation network in two stages, which was successfully utilized to overcome the impact of rainwater on sensors during automatic driving. Object detection, as one of the high-level vision tasks, was also used to guide image super-resolution reconstruction. In the research of [34], the authors built a novel end-to-end super-resolution framework, in which the updating of the super-resolution network was influenced by the detection loss. The proposed task-driven super-resolution algorithm was verified to improve the detection accuracy on the low-resolution image under various scale factors or conditions.

### 3.1. Detection-Driven Fusion Framework

By focusing on landmine detection, we introduce a joint training framework based on detection-driven fusion (DDF), as shown in Figure 6, in which we select YOLOv5 [30] for detection.

The image fusion network is not limited to the preservation of low-level semantic information but is driven by the detection task to gather more valuable high-level semantic information. Meanwhile, the fusion network is partially influenced by the detection result, which can preserve fine-grained details while providing more semantic information for computer perception.

### 3.2. Fusion Network

On the basis of auto-encoder, we design a lightweight fusion network (denoted as DDFNet) with two inputs and one output, in which the registered images include the visible image $I_{rgb}\in R_{H*W*3}$ and the near-infrared image $I_{nir}\in R_{H*W*1}$ as the original inputs. The images pass through the encoder, fusion layer, and decoder, respectively, as shown in Figure 7, where the encoder is used for feature extraction and the decoder for image reconstruction, which can be expressed as follows:(1)$$F_{rgb},F_{nir}=\mathrm{EN}\left(I_{rgb}\right),\mathrm{EN}\left(I_{nir}\right).$$
(2)$$F_{f}=\mathrm{Conv}\left(\mathrm{Concat}\left(F_{rgb},F_{nir}\right)\right).$$
(3)$$I_{f}=\mathrm{DE}\left(F_{f}\right)$$
where $F_{rgb}$ and $F_{nir}$ are the depth features extracted by two different encoders EN, respectively. $F_{f}$ denotes the fused features of the visible and the infrared images through concatenation, and $I_{f}$ is the fused image reconstructed by decoder DE. In Figure 7, the encoder part is mainly composed of two general convolutional blocks as well as two dense residual blocks (DRB), as shown in Figure 8, which refers to the combination of the residual and dense blocks from [35]. The main branch of DRB aggregates shallow and deep semantic information by dense connection as the shortcut branch uses the classic 1×1 convolutional layer in the residual block to achieve identity mapping. Next, the output of the main branch and the shortcut are added at the element level to integrate deep features with fine-grained features. Thus, the superposition of two DRBs in the encoder ensures that the abundant semantic features are fully utilized. Then, the fusion layer fuses the rich features extracted from the infrared image and the visible image via concatenation. Finally, the fused features are fed back into the decoder for image reconstruction. With the dimensions of the feature map shrinking, the decoder outputs a three-channel fused image. In general, the design of our fusion network is simple and lightweight, which contributes to a higher fusion speed, and it is easy to facilitate subsequent deployments. Moreover, the introduction of the auto-encoder architecture as well as the DRB makes it possible for the fused image to fully restore the details and semantic information of the source images.

### 3.3. Loss Function

The design of the loss function starts with two aspects. Firstly, the fusion network needs to fully integrate the complementary information of the source images, such as the salient objects in the infrared image and the texture details in the visible image. For this reason, the content loss $L_{content}$ is constructed, which aims to ensure the visual fidelity of the fused image. Secondly, the fused image should effectively promote the detection effect. Therefore, the semantic loss $L_{detect}$ is introduced to indicate the contribution of the fused image to the detection task. In summary, the joint loss function is designed as follows:(4)$$L_{all}=L_{content}+\lambda L_{detect}.$$
where the joint loss $L_{all}$ of the fusion network comprises the content loss $L_{content}$ and the semantic loss $L_{detect}$, and λ is set as the weight of the semantic loss.

Meanwhile, the content loss is composed of three types of loss, image intensity loss ($L_{intensity}$), texture loss($L_{texture}$) [35], and structural similarity loss ($L_{ssim}$), formatted as
(5)$$L_{content}=L_{intensity}+\beta_{1}L_{texture}+\beta_{2}L_{ssim}.$$
where $\beta_{1}$ and $\beta_{2}$ are the hyperparameters to define the weights of texture loss and structural similarity loss, respectively. Image intensity loss $L_{intensity}$ assesses the difference in pixel intensity between a fused image and a source image, which can be more precisely formulated as
(6)$$L_{intensity}=\frac{1}{HW}{∥I_{f}-\max\left(I_{rgb},I_{nir}\right)∥}_{1}$$
where *H* and *W* are, respectively, the height and width of the fused image $I_{f}$. ${∥\cdot ∥}_{1}$ stands for the $l_{1}$-norm. max(·) ensures that the fused image maintains the intensity sufficiently, which contributes to the positioning of landmines. The texture loss $L_{texture}$ is introduced to preserve the details. It can be calculated as
(7)$$L_{texture}=\frac{1}{HW}{∥\left|\nabla I_{f}\right|-\max\left(\left|\nabla I_{rgb}\right|,\left|\nabla I_{nir}\right|\right)∥}_{1}$$
where the absolute value operation is denoted by |·| and the Sobel gradient operator is denoted by ∇. Although the standard $L_{2}$ distance can maintain edges well in image fusion, it cannot effectively remove noise. Moreover, it lacks the capability to determine how structurally similar the images are, and it is highly sensitive to changes in lighting, color, etc., which poses a challenge in maintaining specificity in multispectral images. The Structure Similarity Index Measure (SSIM) proposed by [36] is used to compare the structural similarity of an image before and after alteration. SSIM is sensitive to local structural changes, as with the human visual system (HVS). More importantly, SSIM can better tolerate the influence of outside information on the source image. The SSIM between two images can be more precisely expressed as follows:(8)$$SSIM_{x,y}=\sum_{x_{i},y_{i}}\frac{2\mu_{x_{i}}\mu_{y_{i}}+C_{1}}{\mu_{x_{i}}^{2}+\mu_{y_{i}}^{2}+C_{1}}\times \frac{2\sigma_{x_{i}}\sigma_{y_{i}}+C_{2}}{\sigma_{x_{i}}^{2}+\sigma_{y_{i}}^{2}+C_{2}}\times \frac{\sigma_{x_{i}y_{i}}+C_{3}}{\sigma_{x_{i}}\sigma_{y_{i}}+C_{3}}$$
where μ and σ represent the mean and the covariance of images, respectively, and $C_{1}$, $C_{2}$, $C_{3}$ are the parameters used to stabilize the measurement. Therefore, the SSIM loss $L_{ssim}$ can be formulated as
(9)$$L_{ssim}=w_{1}\left(1-SSIM_{I_{f},I_{rgb}}\right)+w_{2}\left(1-SSIM_{I_{f},I_{nir}}\right)$$
where $w_{1}$ and $w_{2}$ denote the weights of structural loss between the fused image $I_{f}$ and the source image $I_{rgb}$ and $I_{nir}$, respectively. Although content loss can maintain information in the fused image and minimize distortion to some extent, it is challenging to measure the promotion impact of the fusion result on high-level vision tasks. As a consequence, we redefine the semantic loss using the standard loss function based on YOLOv5 [30] to guide the training of the fusion network and ultimately enhance the detection performance.

The object detection task normally requires us to refine the target size as well as identify and categorize the detected item. In particular, the accuracy of the test results depends on three optimization directions:All targets in the image should be identified, with a low missed and false alarm rate;The bounding boxes should completely and accurately enclose the target;The categorization of the detected object should be consistent with its label.

Therefore, the semantic loss (marked $L_{detect}$ [30]) is defined by the confidence loss (marked $loss_{conf}$), the bounding box loss (marked $loss_{rect}$), and the categorized loss (marked $loss_{cls}$) as follows:(10)$$L_{detect}=a\times loss_{rect}+b\times loss_{conf}+c\times loss_{cls}.$$
where *a*, *b*, and *c* are their weights, respectively. The confidence loss generally weighs the most, followed by the bounding box loss and the categorized loss.

### 3.4. Joint Training Algorithm

There are primarily two approaches to low-level vision model training that is motivated by high-level visual tasks. The first one is to train the low-level vision model under the guidance of the pre-trained high-level vision model, but the training result cannot be effectively fed back to the high-level vision model. The second one is to concurrently train the low-level and high-level vision models in one stage, and this strategy may bring difficulties in maintaining a balance between the performance of vision models at various levels. Therefore, we introduce a joint training method based on an iterative strategy to train our fusion network.

More specifically, we train the fusion and detection networks repeatedly, with the number of iterations set to *n*. Firstly, the fusion and the detection network employ the Adaptive Moment Estimation (ADAM) optimizer to maximize each of their individual parameters under the supervision of the joint loss $L_{all}$ and the detection loss $L_{detect}$, respectively. Secondly, the parameters of the detection model are fixed to generate real-time semantic loss when the fusion network is updated. It should be noted that the hyperparameter λ set for the joint loss is dynamically adjusted in an iterative process, which is expressed as follows:(11)$$\lambda_{n}=n\times \lambda_{n-1}$$

This option is primarily intended to guarantee that, as the fusion network has been trained to a certain extent, the fraction of semantic loss steadily increases, and the visual effect becomes a secondary priority.

During the training process, we adopt the periodic joint training strategy demonstrated in Algorithm 1, which is iterated for 5 rounds.
**Algorithm 1:** Multi-stage joint training algorithm.
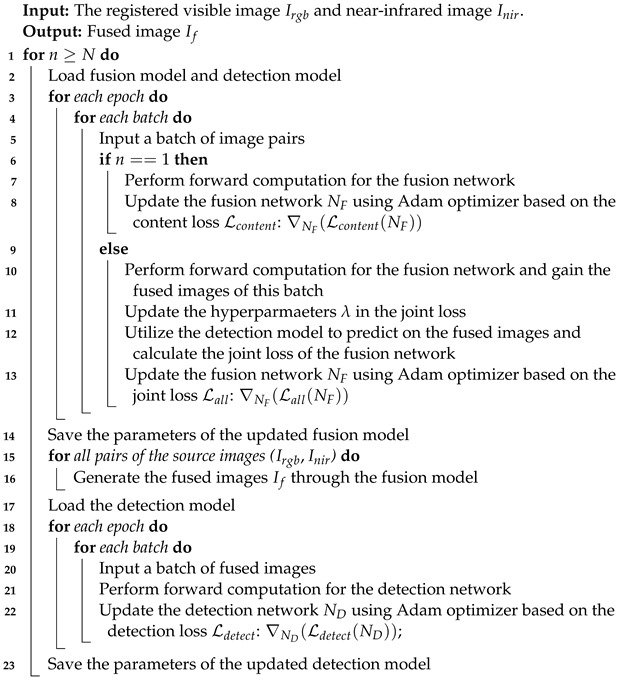


In the first round, the fusion network only utilizes the content loss for backward propagation, while the semantic loss is set to 0. Then, in the next four rounds, both the content and semantic loss contribute to the updating of the fusion network. In addition, the detection network is trained through the updated fused image in each round.

## 4. Experiment and Results

### 4.1. Fusion Result

#### 4.1.1. Comparative Experiment

In our experiments, the landmine dataset is randomly divided into a training part and a testing part at a ratio of 8:2. The hyperparameters involved are all set to the default values, with no validation set for adjustment. This experimental setup makes it easier to demonstrate the feasibility of our landmine detection method.

To begin with, we conducted fusion comparative experiments for DDF and other algorithms. Figure 9 shows fused results of the selected deep learning fusion models and some conventional algorithms in the Visible and Infrared Image Fusion Benchmark [37,38] (VIFB). It should be noted that since FusionGAN [39] and RFN-Nest [40] only work for single-channel images with a depth of 8 bits, the fusion result is also a single-channel grayscale image. The testing images are randomly selected from the infrared and visible image pairings from the multispectral landmine dataset, and part of the region is intercepted to illustrate the fusion effect. There are two types of scatterable mines (M93 and T125) in the interception area, of which the rose-red mine (M93) is easy to identify due to its great size and distinguishing characteristics. T125 is partially covered by vegetation, and therefore it is difficult to detect in visible images but it reveals strong visual features in near-infrared images. It can be observed from Figure 9 that M93 is still clearly identifiable in the fusion results of most algorithms, while the distortion is severe in certain fused images. For example, testing images employing Densefuse [41], IFCNN [42], LatLRR [43], and GFCE [44] perform poorly in terms of pixel intensity and color. Moreover, the fused image generated by FusionGAN has the lowest image quality, and the target characteristics are nearly lost. Compared to the source images (RGB and NIR), RFN-Nest [40] could successfully integrate the complementary features of the two spectra. However, although the significant features of landmines in the two images are likewise preserved in the fused image, the color information is lost and is visibly different. In contrast, U2Fusion [45] well maintains the color feature distribution, but it introduces some noise in the white area. As for MST-SR [46], NSCT-SR [46], and the DDF that we propose, all of them enhance the visual saliency of T125 while preserving the original color.

Next, we randomly chose 20 pairs of near-infrared and visible images from the multispectral landmine dataset and processed them using different fusion algorithms. We performed a comprehensive fusion evaluation using 13 statistical metrics [37,38], which is shown in Table 2.

To some extent, the evaluation metrics are designed to quantitatively simulate the human visual system (HVS). Some of them, such as the Root Mean Square Error (RMSE), Peak Signal to Noise Ratio (PSNR), and Structural Similarity Index Measure (SSIM), are measured based on the original RGB and NIR images as benchmarks, which can reflect the amount of complementary information retained from the source images in the fused image. Others, such as the Average Gradient (AG), Entropy Index (EI), and Entropy (EN), can be calculated with the fused image itself and be used for the evaluation of the image quality, such as edges, saliency, and other features, which may be helpful for the subsequent detection.

In Table 2, blue represents the best-performing model for the corresponding metric. For the quantitative metrics designated as ↓, a lower value is better. The last row of Table 2 displays the evaluation metrics of DDF, where red, orange, and green, respectively, represent excellent, moderate, and poor performance of the fused image based on DDF compared with other fusion algorithms. It can be seen that LatLRR [43] ranks the highest for approximately half of the metrics, but it is not satisfactory regarding the visual effect shown in the aforementioned Figure 9. NSCT-SR [46] and MST-SR [46] perform better in the metrics of $Q_{CB}$ [48] and $Q_{CV}$ [49], as it can be seen in Figure 9 that both of them provide fusion results consistent with human vision. The DDF that we propose demonstrates satisfactory performance in most of the statistical indicators, while only providing unsatisfactory assessment results for Cross-Entropy (CE) and the Peak Signal to Noise Ratio (PSNR).

#### 4.1.2. Generalization Experiment

In addition, in order to verify the generalization performance of the proposed method, we also chose 21 pairs of infrared and visible images from VIFB [37,38] as a further evaluation. Figure 10 offers a qualitative comparison between DDF and the other 10 fusion models. It can be observed that the fusion results of Densefuse [41], U2Fusion [45], FusionGAN [39], RFN-Nest [40], GFCE [44], MST-SR [46], and NSCT-SR [46] are significantly affected by radiation or lose a lot of color information; among them, RFN-Nest [40] shows blurred edges and a loss of detail. IFCNN [42] and DDF achieve satisfactory fusion results, presenting excellent visual effects. For example, the textural details of the distant buildings in the image are greatly enhanced compared with the original visible image.

Table 3 lists the mean values of the 21 pairs of image fusion metrics for the VIFB [37,38] dataset. It can be observed that the performance of DDF is mostly moderate. The reason is that the DDF is trained with the scatterable mine images—more specifically, the near-infrared and visible images—while the infrared images in VIFB [37,38] are mainly thermal infrared. The contributions of the different statistical metrics to the final advanced visual task are still difficult to analyze and determine, but in the experimental study of this section, DDF shows better performance in the evaluation of some metrics and has an excellent visual effect.

### 4.2. Detection Performance

Furthermore, we conducted comparisons of the detection performance for the aforementioned fusion models. During the training and testing process, we employed two NVIDIA GeForce RTX 3090 graphics cards. We randomly shuffled and divided the collected data into a training set and a testing set at a ratio of 8:2, with quantities of 3964 and 991, respectively.

#### 4.2.1. Landmine Detection Evaluation

To further validate the DDF that we propose under the condition of vegetation, we compared the detection performance of DDF with other fusion methods. We randomly selected an image pair from the testing set for landmine detection. Figure 11 shows the detection results based on different fusion methods. Except for the source images RGB and NIR, the first two rows are based on deep learning fusion methods, while the last row relates to conventional fusion methods. As can be seen from the source images, most of the landmines in RGB can be distinguished more clearly. However, a small number of landmines, such as T125 and M93, magnified to be displayed in ground truth, show clearer outlines in the NIR image, which demonstrates the complementary information of RGB and NIR images. Therefore, it can be observed that most of the landmines, besides the aforementioned T125, can be directly detected in RGB, outperforming most of the fusion-based algorithms. DDF successfully detects all landmines, even when some of them are covered with vegetation. It is worth noting that the detection of T125 is not so precise, as the main body of its long strip is obscured by weeds. MST-SR [46] can also obtain a high recall rate, but it may cause false alarms. NSCT-SR [46] is able to detect most of the landmines, with the exception of B91, which is occluded severely.

Then, we produced a statistical table according to the detection results of all the images in the testing set. The detection evaluation metrics of various fusion models are shown in Table 4, in which red and blue, respectively, represent the best result and the second-best result. It can be observed that the DDF has the best performance in terms of precision, recall, and mAP@0.5. In terms of the recall rate, the detection results of most fusion models are even worse than those of single-spectrum visible images, which also indicates that the existing fusion algorithms do not focus sufficiently on high-level visual tasks. The superior performance of DDF demonstrates the validity and feasibility of our methodology.

#### 4.2.2. Efficiency Evaluation

Next, we evaluated the inference efficiency of different fusion models, which is shown in Table 5. The experiment was conducted twice, one on 20 pairs of randomly selected landmine images and the other on the VIFB [37,38] dataset, including 21 pairs of visible and infrared images. Only one NVIDIA GeForce RTX 3090 graphics card was utilized for inference. Table 5 shows the time consumption of the different models—more specifically, the average time consumption of the fusion process for each pair of images. The calculation covers the process of data loading and image storage, while the time for model loading is not included. Since the efficiency of the traditional algorithm is generally much lower than that of the deep learning model with the forward computation, the table is separated into two parts: fusion models based on either the traditional or deep learning methods. Green highlights the highest time cost in the current column, while red indicates the lowest. On one hand, we can observe that the well-performing algorithm LatLRR [43], based on the traditional method, is much more time-consuming than the others. On the other hand, except for CNN [52] used as a reference, the distinction between the models based on deep learning is not obvious. Meanwhile, the inference efficiency of our model is moderate, whereas the IFCNN [42] model, which performs poorly in terms of the visual effect or the detection task, shows slight advantages in inference efficiency.

## 5. Conclusions

In recent years, the application of deep-learning-based techniques in landmine detection has progressed at a rapid pace. This paper investigates recent advances and suggests some opportunities for future research in this field. Specifically, we propose a UAV-based solution for the detection of scatterable landmines. We establish a dataset of scatterable landmines, including RGB and NIR image pairs collected by a multispectral sensor, with complex scenes covered. In addition, to fully utilize the multispectral potential in detection, we employ a training framework for image fusion, which is task-oriented and allows high-level semantic information to flow back into the fusion network upstream, so as to boost the quality of the fusion image as well as enhance the detection performance purposefully. We have carried out extensive comparative experiments to verify the effectiveness of the fusion and detection methods that we propose. Our attempt at multispectral utilization for landmine detection paves the way for future advancement.

Overall, the obscuration and burial of landmines remain the most challenging problems that we confront. We have focused on the concealment and occlusion of landmines by vegetation when constructing the dataset. However, the simulated settings are still far from the natural minefield environment. The aging, corrosion, and burial of mines in real situations are time-consuming and complex to reproduce. Furthermore, the range of host conditions, such as snow, gravel pavement, and so on, needs to be greatly enlarged. In the future, we will supplement the dataset with more hard samples (primarily obscured by weeds, fallen leaves, sand, snow, and gravel or buried under shallow soils). We expect that the proposed dataset will encourage the development of better landmine detection methods, and our efforts will be of assistance and inspiration to demining organizations.

## Figures and Tables

**Figure 1 sensors-23-05693-f001:**
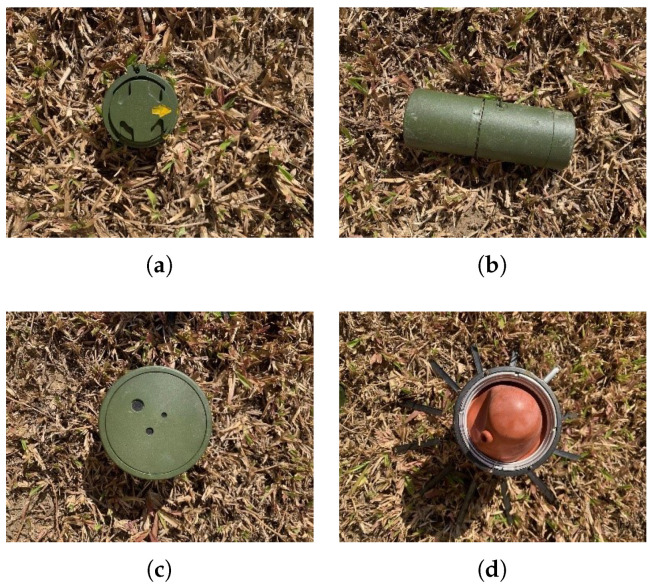
Landmine samples. (**a**) Type: M14; size: diameter 58 mm × height 38 mm; (**b**) Type: T125; size: diameter 50 mm × height 135 mm; (**c**) Type: B91; size: diameter 147 mm × height 66 mm; (**d**) Type: M93; size: diameter 122 mm × height 168 mm.

**Figure 2 sensors-23-05693-f002:**
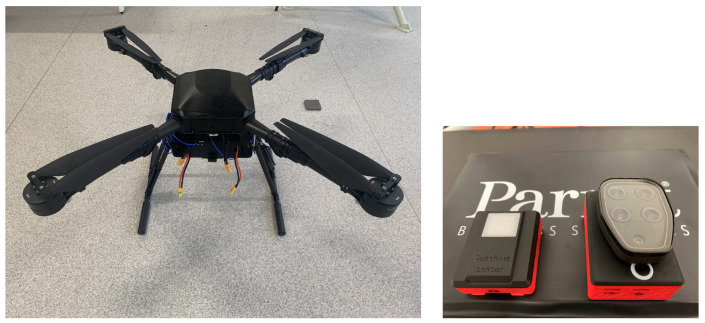
Equipment: **Left**—UAV; **Right**—Sequoia kit (with a sunshine sensor and a multispectral camera).

**Figure 3 sensors-23-05693-f003:**
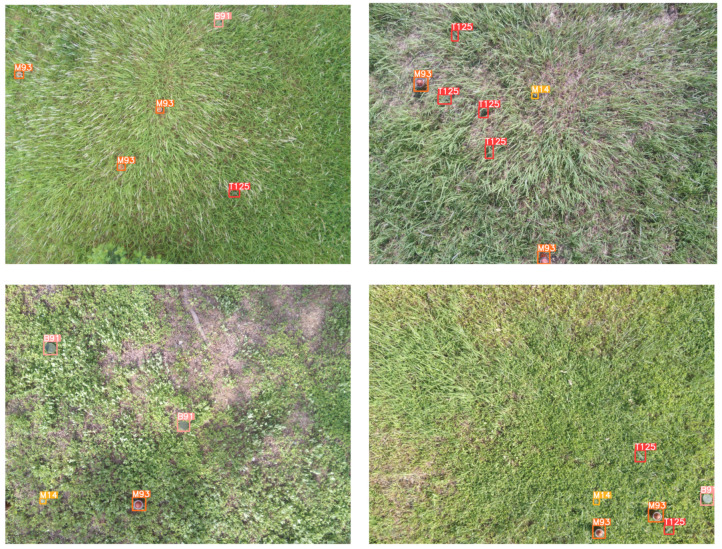
Mines on different backgrounds.

**Figure 4 sensors-23-05693-f004:**
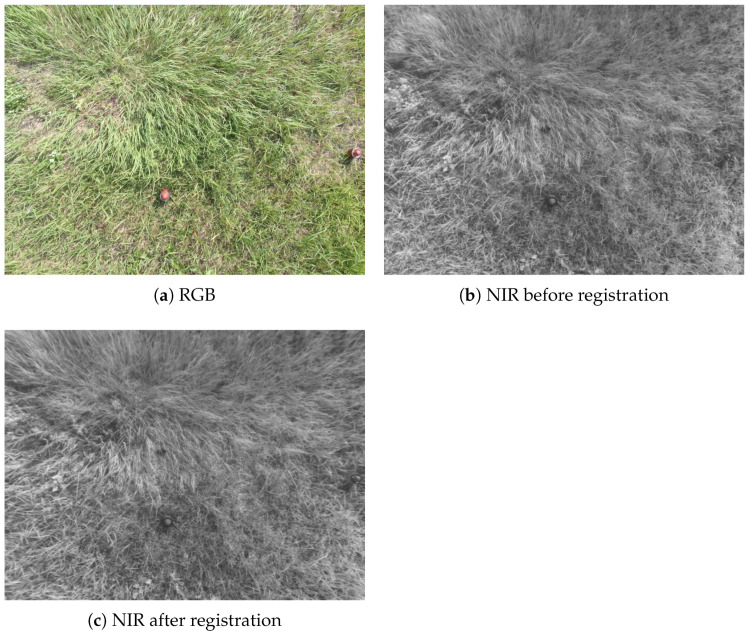
Registration, showing that some mines are clearer in NIR.

**Figure 5 sensors-23-05693-f005:**
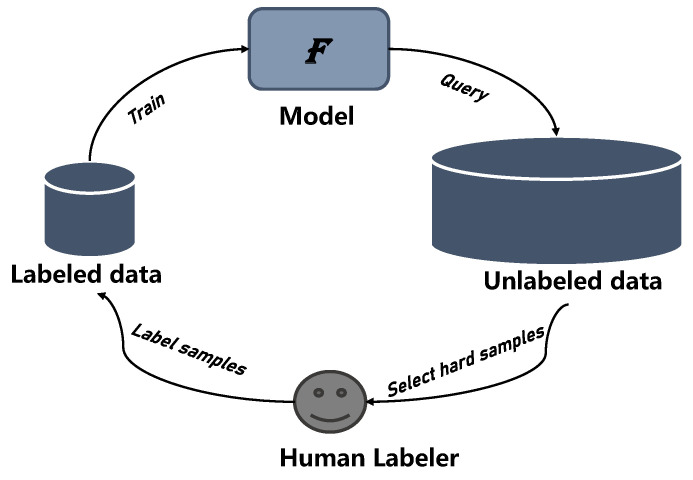
The workflow of active learning.

**Figure 6 sensors-23-05693-f006:**
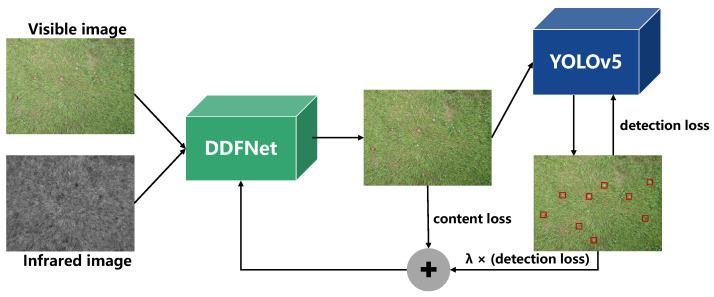
The architecture of the joint training framework based on detection-driven fusion (DDF).

**Figure 7 sensors-23-05693-f007:**
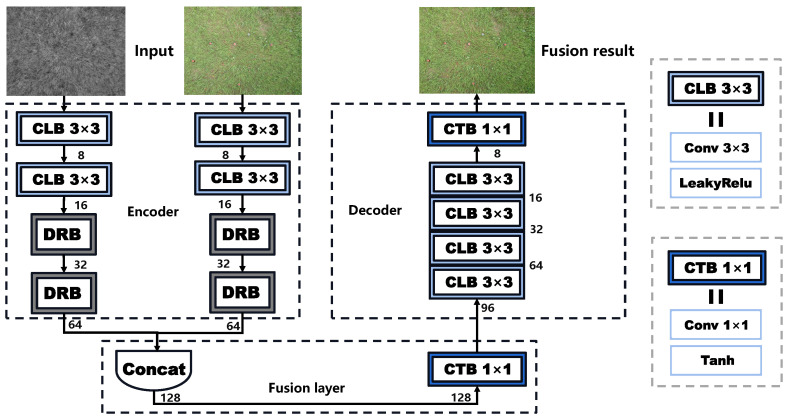
The architecture of the visible and infrared image fusion network (termed DDFNet), which is based on auto-encoder. The convolution leaky ReLu block (CLB 3×3) is composed of a 3×3 convolutional layer and a leaky ReLu layer. The convolution tanh block (CTB 1×1) is composed of a 1×1 convolutional layer and a tanh layer.

**Figure 8 sensors-23-05693-f008:**
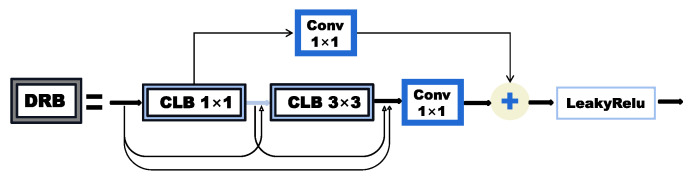
The structure of the dense residual block (DRB). Convolution leaky ReLu block (CLB) is composed of a convolutional layer and a leaky ReLu layer.

**Figure 9 sensors-23-05693-f009:**
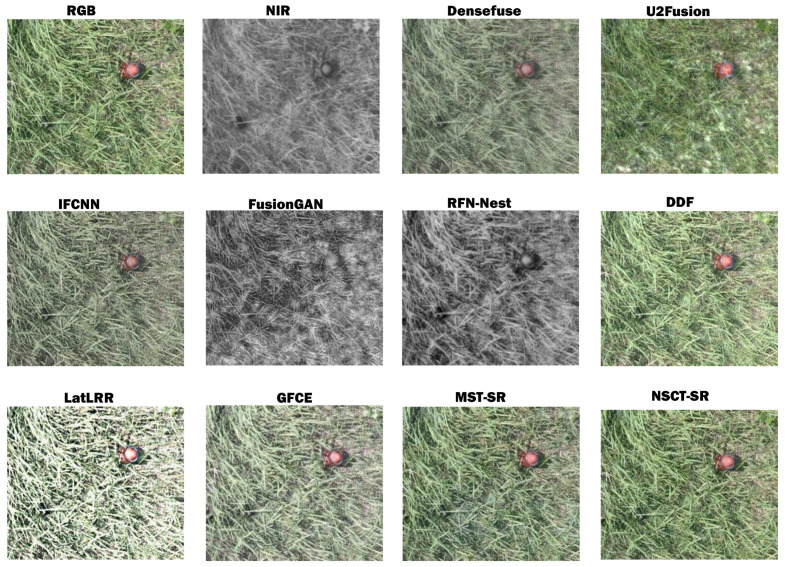
Qualitative comparisons of fusion results for a visible (RGB) and near-infrared (NIR) image pair from landmine dataset.

**Figure 10 sensors-23-05693-f010:**
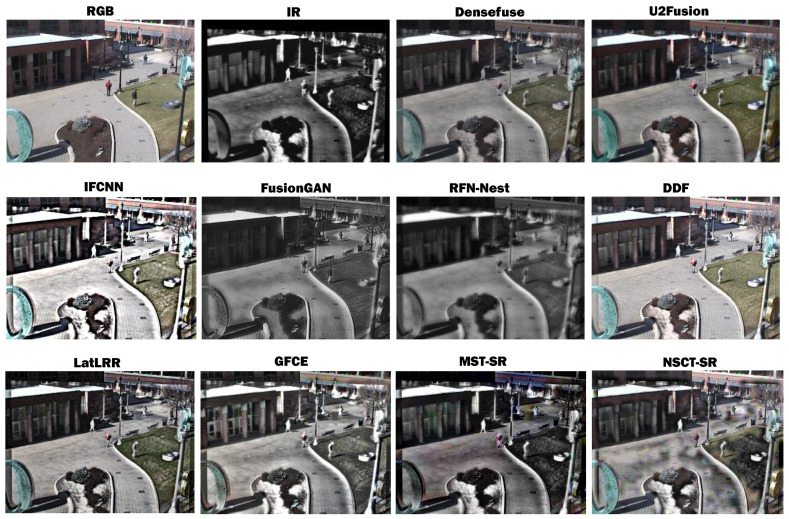
Qualitative comparisons of fusion results for a visible (RGB) and near-infrared (NIR) image pair obtained from [37,38].

**Figure 11 sensors-23-05693-f011:**
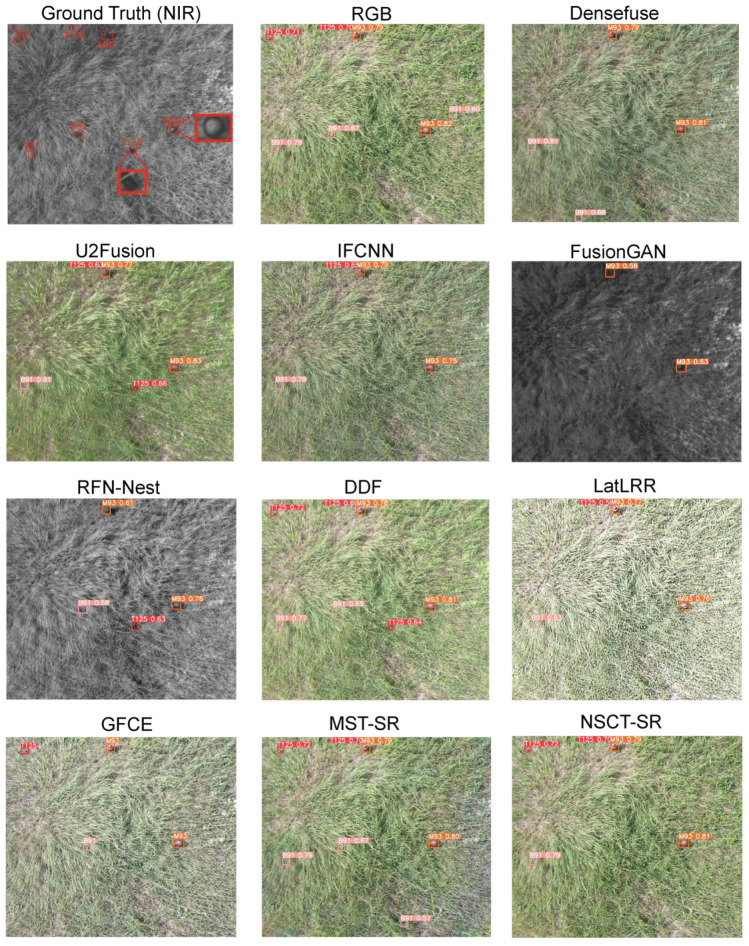
Qualitative comparisons of detection performance based on different fusion models and the same detection model (YOLOv5 [30]). The source image NIR is not considered in the comparison as it lacks quantities of color information for detection, so it is shown as ground truth.

**Table 1 sensors-23-05693-t001:** Details of Sequoia bands.

Band Name	Wavelength	Bandwidth	Definition
Green	550 nm	40 nm	1.2 Mpx
Red	660 nm	40 nm	1.2 Mpx
Red-edge	735 nm	10 nm	1.2 Mpx
Near-infrared	790 nm	40 nm	1.2 Mpx
RGB			16 Mpx

**Table 2 sensors-23-05693-t002:** Quantitative comparisons of 13 metrics [37,38], i.e., Average Gradient (AG), Cross-Entropy (CE), Entropy Index (EI), Entropy (EN), Mutual Information (MI), Peak Signal to Noise Ratio (PSNR), $Q^{AB/F}$ [47], $Q_{CB}$ [48], $Q_{CV}$ [49], Root Mean Square Error (RMSE), Spatial Frequency (SF), Structural Similarity Index Measure (SSIM), and Standard Deviation (SD), on 20 pairs of near-infrared and visible images from the landmine dataset. Blue indicates the best result. Red, orange, and green represent excellent, moderate, and poor performance, respectively, of the fused image based on DDF compared with other fusion algorithms. ↓ indicates that a lower result is better; otherwise, a higher value is better.

	AG	CE↓	EI	EN	MI	PSNR	$Q^{\mathrm{AB}/F}$ [47]	$Q_{\mathrm{CB}}$ [48]	$Q_{\mathrm{CV}}$↓ [49]	RMSE↓	SF	SSIM	SD
NIR	7.77	0.38	76.85	6.73	4.72	63.36	0.36	0.76	258.12	**0.03**	16.84	1.16	26.32
RGB	19.19	0.25	172.83	7.35	5.15	63.36	0.72	0.67	50.26	**0.03**	46.56	1.16	39.91
ADF [50]	15.33	0.23	135.17	6.96	0.94	62.98	0.55	0.53	67.19	**0.03**	37.07	1.14	30.18
CBF [51]	17.43	0.22	157.78	7.23	1.41	62.87	0.60	0.58	** 44.02 **	**0.03**	42.17	1.15	36.40
CNN [52]	19.19	0.26	173.17	7.35	1.19	62.52	0.62	0.60	49.31	0.04	46.29	1.13	39.88
DLF [53]	11.57	0.32	107.21	6.79	0.92	63.20	0.39	0.56	67.18	**0.03**	27.25	1.24	26.83
FPDE [54]	10.93	0.36	102.27	6.74	0.90	63.21	0.36	0.56	69.68	**0.03**	25.23	1.24	25.98
GFCE [44]	19.74	0.52	178.80	7.37	1.05	61.36	0.51	0.57	54.73	0.05	47.53	1.19	40.43
GFF [55]	19.13	0.26	173.62	7.35	**1.56**	62.64	0.63	0.62	50.90	0.04	45.82	1.11	39.88
IFEVIP [56]	14.95	0.96	139.82	7.23	1.06	60.55	0.39	0.57	188.01	0.06	34.99	1.21	37.23
LatLRR [43]	**32.01**	1.07	**92.68**	**7.50**	1.03	59.11	0.38	0.59	142.03	0.08	**77.57**	0.94	**62.47**
MGFF [57]	18.80	0.19	173.10	7.41	0.95	62.48	0.47	** 0.61 **	100.80	0.04	44.61	1.23	41.67
MST_SR [46]	19.16	0.21	173.35	7.35	1.26	62.58	0.62	** 0.61 **	48.88	0.04	46.05	1.13	39.68
MSVD [58]	15.00	0.20	132.86	7.00	0.82	62.88	0.42	0.50	72.12	**0.03**	39.02	1.17	30.98
NSCT_SR [46]	19.13	0.26	173.36	7.35	1.55	62.64	**0.63**	** 0.61 **	49.99	0.04	45.91	1.12	39.84
ResNet [59]	10.72	0.36	100.72	6.73	0.90	63.20	0.34	0.56	71.61	**0.03**	24.72	1.25	25.83
RP_SR [46]	18.84	0.24	167.85	7.32	1.12	62.43	0.58	0.56	60.42	0.04	46.31	1.14	38.92
TIF [60]	15.57	0.17	144.09	7.18	0.93	62.74	0.46	0.60	60.56	**0.03**	37.35	1.18	35.25
VSMWLS [61]	18.93	**0.16**	164.55	7.23	0.93	62.70	0.51	0.55	66.87	**0.03**	47.41	1.19	36.70
Densefuse [41]	10.97	0.35	101.20	6.74	1.00	**63.32**	0.39	0.57	68.92	**0.03**	26.05	** 1.30 **	26.01
IFCNN [42]	18.14	0.17	161.98	7.18	1.06	62.97	0.55	0.53	68.25	**0.03**	44.46	1.27	35.22
FusionGAN [39]	15.26	0.23	136.93	7.26	0.25	60.88	0.44	0.43	485.24	0.05	37.39	0.99	37.40
RFN-Nest [40]	12.11	0.25	119.23	7.11	0.89	62.71	0.48	0.59	97.02	0.04	25.75	1.22	33.52
U2Fusion [45]	12.12	0.25	114.42	7.00	0.33	61.30	0.29	0.46	337.34	0.05	26.74	0.89	31.53
DDF	** 18.92 **	** 0.44 **	** 173.87 **	** 7.33 **	** 1.45 **	** 61.96 **	** 0.62 **	**0.58**	** 45.14 **	**0.04**	**44.53**	** 1.20 **	**39.31**

**Table 3 sensors-23-05693-t003:** Quantitative comparisons of 13 metrics [37,38], i.e., Average Gradient (AG), Cross-Entropy (CE), Entropy Index (EI), Entropy (EN), Mutual Information (MI), Peak Signal to Noise Ratio (PSNR), $Q^{AB/F}$ [47], $Q_{CB}$ [48], $Q_{CV}$ [49], Root Mean Square Error (RMSE), Spatial Frequency (SF), Structural Similarity Index Measure (SSIM), and Standard Deviation (SD), on 21 pairs of near-infrared and visible images from the VIFB [37,38] dataset. Blue indicates the best result. Red, orange, and green represent excellent, moderate, and poor performance, respectively, of the fused image based on DDF compared with other fusion algorithms. ↓ indicates that a lower result is better; otherwise, a higher value is better.

	AG	CE↓	EI	EN	MI	PSNR	$Q^{\mathrm{AB}/F}$ [47]	$Q_{\mathrm{CB}}$ [48]	$Q_{\mathrm{CV}}$↓ [49]	RMSE↓	SF	SSIM	SD
ADF [50]	4.58	1.46	46.53	6.79	1.92	58.41	0.52	0.47	777.82	**0.10**	14.13	1.40	35.19
CBF [51]	7.15	0.99	74.59	7.32	2.16	57.59	0.58	0.53	1575.15	0.13	20.38	1.17	48.54
CNN [52]	5.81	1.03	60.24	7.32	2.65	57.93	**0.66**	0.62	512.57	0.12	18.81	1.39	** 60.08 **
DLF [53]	3.82	1.41	38.57	6.72	2.03	**58.44**	0.43	0.45	759.81	**0.10**	12.49	**1.46**	34.72
FPDE [54]	4.54	1.37	46.02	6.77	1.92	58.40	0.48	0.46	780.11	**0.10**	13.47	1.39	34.93
GFCE [44]	7.50	1.93	77.47	7.27	1.84	55.94	0.47	0.53	898.95	0.17	22.46	1.13	51.56
GFF [55]	5.33	1.19	55.20	7.21	2.64	58.10	0.62	0.62	881.62	0.11	17.27	1.40	50.06
GTF [62]	4.30	1.29	43.66	6.51	1.99	57.86	0.44	0.41	2138.37	0.12	14.74	1.37	35.13
HMSD_GF [63]	6.25	1.16	65.03	7.27	2.47	57.94	0.62	0.60	532.96	0.12	19.90	1.39	57.62
Hybrid_MSD [63]	6.13	1.26	63.49	7.30	2.62	58.17	0.64	0.62	510.87	0.11	19.66	1.41	54.92
IFEVIP [56]	4.98	1.34	51.78	6.94	2.25	57.17	0.49	0.46	573.77	0.14	15.85	1.39	48.49
LatLRR [43]	**8.96**	1.68	**92.81**	6.91	1.65	56.18	0.44	0.50	697.29	0.17	** 29.54 **	1.18	57.13
MGFF [57]	5.84	1.29	60.61	7.11	1.77	58.21	0.57	0.54	676.89	0.11	17.92	1.41	44.29
MST_SR [46]	5.85	0.96	60.78	7.34	2.81	57.95	**0.66**	**0.64**	522.69	0.12	18.81	1.39	57.31
MSVD [58]	3.54	1.46	36.20	6.71	1.95	58.41	0.33	0.43	808.99	**0.10**	12.53	1.43	34.37
NSCT_SR [46]	6.49	** 0.90 **	67.96	** 7.40 **	**2.99**	57.43	0.65	0.62	1447.34	0.13	19.39	1.28	52.47
ResNet [59]	3.67	1.36	37.26	6.73	1.99	**58.44**	0.41	0.44	724.83	**0.10**	11.74	**1.46**	34.94
RP_SR [46]	6.36	0.99	65.22	7.35	2.34	57.78	0.57	0.61	888.85	0.12	21.17	1.33	55.81
TIF [60]	5.56	1.37	57.84	7.08	1.77	58.23	0.58	0.54	613.00	0.11	17.74	1.40	42.64
VSMWLS [61]	5.61	1.41	57.25	7.03	2.03	58.19	0.55	0.50	754.70	0.11	17.66	1.42	46.25
Densefuse-L1 [41]	3.54	1.34	36.17	6.70	2.03	**58.44**	0.37	0.44	762.80	**0.10**	11.02	**1.46**	34.24
Densefuse-Add [41]	3.54	1.34	36.17	6.70	2.03	**58.44**	0.37	0.44	762.80	**0.10**	11.02	**1.46**	34.24
IFCNN-Max [42]	5.85	1.56	60.39	6.91	2.00	58.04	0.58	0.47	**470.48**	0.11	18.67	1.41	44.15
IFCNN-Sum [42]	5.32	1.62	54.43	6.84	1.92	58.39	0.57	0.47	757.32	**0.10**	17.61	1.45	36.99
IFCNN-Mean [42]	5.03	1.64	50.96	6.77	1.91	58.39	0.53	0.45	742.89	**0.10**	16.64	1.45	36.37
FusionGAN [39]	4.25	1.56	43.51	6.77	1.92	57.97	0.48	0.47	825.04	0.11	14.56	1.38	39.50
RFN-Nest [40]	3.66	1.50	39.42	7.15	2.08	58.10	0.41	0.48	829.63	0.11	10.03	1.40	45.36
U2Fusion [45]	3.57	1.12	38.00	6.92	2.14	58.28	0.41	0.50	739.27	0.11	10.16	**1.46**	40.26
DDF	**5.63**	** 1.52 **	**58.61**	** 6.96 **	**2.12**	** 57.28 **	**0.59**	** 0.46 **	** 417.23 **	** 0.13 **	**17.64**	**1.45**	** 49.40 **

**Table 4 sensors-23-05693-t004:** Quantitative comparisons of detection performance based on different fusion models and the same detection model (YOLOv5 [30]). Red indicates the best result. Blue represents the second-best result.

All					
	**TP**	**FP**	**Precision**	**Recall**	**mAP@0.5**
Labels	991	\	\	\	\
NIR [29]	674	229	0.746	0.680	0.632
RGB [29]	840	123	0.872	0.848	0.848
Decision Fuse [29]	** 859 **	196	0.814	**0.867**	0.841
DDF	** 898 **	** 64 **	** 0.933 **	** 0.906 **	** 0.922 **
IFCNN [42]	835	93	0.9	0.843	** 0.884 **
Densefuse [41]	856	84	** 0.911 **	0.864	** 0.884 **
U2Fusion [45]	833	84	0.908	0.841	0.87
RFN-Nest [40]	713	134	0.842	0.719	0.728
FusionGAN [39]	585	** 59 **	0.908	0.590	0.611

**Table 5 sensors-23-05693-t005:** Time consumption of 25 typical fusion models (Unit: seconds per image pair). Red indicates the fastest. Green indicates the slowest.

Conventional Fusion Models			Deep-Learning-Based Fusion Models		
	* **VIFB** *	* **Landmine** *		* **VIFB** *	* **Landmine** *
ADF [50]	1.00	2.89	CNN [52]	** 31.76 **	** 117.82 **
CBF [51]	22.97	80.29	DLF [53]	18.62	36.68
FPDE [54]	2.72	10.12	ResNet [59]	4.8	10.76
GFCE [44]	2.13	6.62	Densefuse [41]	0.03	0.51
GFF [55]	0.41	1.02	IFCNN-Max [42]	0.03	0.21
IFEVIP [56]	0.17	** 0.35 **	IFCNN-Sum [42]	** 0.02 **	** 0.20 **
LatLRR [43]	** 271.04 **	** 910.48 **	IFCNN-Mean [42]	0.03	0.24
MGFF [57]	1.08	3.33	FusionGAN [39]	0.38	0.65
MST_SR [46]	0.76	2.20	RFN-Nest [40]	0.08	0.44
MSVD [58]	1.06	2.27	U2Fusion [45]	0.04	0.28
NSCT_SR [46]	94.65	500.29	DDF	0.08	0.24
RP_SR [46]	0.86	2.91			
TIF [60]	** 0.13 **	0.37			
VSMWLS [61]	3.51	12.87			

## Data Availability

Not applicable.

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
