# Peer review of "Joint Fusion and Detection via Deep Learning in UAV-Borne Multispectral Sensing of Scatterable Landmine†"

_sensors, 2023, doi:10.3390/s23125693_

Round 1

Reviewer 1 Report

It was not discussed the influence of types of ground vegetation on the result of detection performance.

formal notes:

·        Figure 1; the symbol for radius is not the Greek capital phi

·        Chapter 2.3; missing the right bracket

·        Figure 7, 8 and 9 are plotted before they are mentioned in the text

·        Eq. 2; weights beta1 and beta2 are not defined

·        Eq. 3; weights a, b and c are not defined

·        Algorithm 1 is printed before it is mentioned in the text

·        Chapter 4.2.2; first sentence, do not use capital w (repeated multiple times)

·        Chapter 5 Conclusion; a multispectral instead of an multispectral

Reviewer 2 Report

A new method for landmine detection is represented in this paper. Authors proposed innovative method which is based on multispectral images acquired from UAV-borne camera and then applies deep learning for the tasks of fusion and detection. This is interesting topic and, as authors have shown, produces promising results. References are appropriate, relevant and well cited.

However, there are some drawbacks that should be addressed.

Figures 3, 4, 5, 11 and 12 are too small. They represent input data and results of the algorithm and should be larger. Details, that are discussed in paragraphs before and after the figures cannot be clearly seen. (In ’Recommendations for Authors’, answering the question ’Are the results clearly presented?’ I checked the box ’Must be improved’ having these figures on my mind, not the explanations or tables).

Page 2, Par. 3 Acronym R-CNN is not explained.

Page 4, Sec. 2.3 Typo: ’(Figure 3. In stead of normally...)’ should be ’(Figure 3. Instead of normally...)’.

Page 6, Par. 1 Acronym CFOG is not explained.

Page 13, caption of Figure 12 Instead of ’Quanlitative’ it should be ’Quantitative’.

Page 14, first paragraph of Conclusion Instead of ’in detection, We employ a...’ it should be ’ in detection, we employ a...’.

Reviewer 3 Report

The paper 'UAV-Borne Landmine Detection via Multispectral Fusion and Deep Learning' proposes a fusion and detection framework for landmine detection. This manuscript lacks many important details. The experiment does not fully validates the article. Here are several suggestions I believe could improve the quality of this manuscript:

1. Compared to such a long introduction section, methods are described so briefly that many important details like math, method benefits, motivations, and comments are missing.

2. DDF should be mentioned in section 3.1. What is CLB? Moreover, please explain the motivation for DDF and why it is better than other fusion networks. The structure presented in Fig.7 is simple and straightforward. I can not distinguish it from other research.

3. All losses in equation (1)(2)(3) should be clear. Since image fusion does not obtain valid ground truth, it is difficult to understand how to measure image fusion quality.

4. Joint-training algorithm is not backed up by any experiment.

5. Most comparing algorithms are not cited and commented.

6. All metrics should be in clear form. I would suggest marking either 'higher the better' or 'lower the better' for readers.

7. What is the meaning of different colors in Table 2 and 3? Please explain them in the captions.

8. I am confused about the RGB-NIR fusion comparisons. Obviously, the authors have labeled fused images as RMSE, PSNR, and SSIM are used. My naive thought is that there should be no ground truth label for this fusion comparison since the 'quality' of the fusion is ad hoc. Please comment on this in the article.

9. For the detection experiment, please provide the number of training and testing data sets and show the difference between them. Qualitative sample results are also beneficial to readers.

Minor issue: 

There are some grammar issues that can be improved in future versions.

There are some grammar issues that can be improved in future versions.
